# Biofortification of Soybean (*Glycine max* L.) through FeSO₄·7H₂O to Enhance Yield, Iron Nutrition and Economic Outcomes in Sandy Loam Soils of India

Salwinder Singh Dhaliwal [1,*], Vivek Sharma [1], Arvind Kumar Shukla [2,*], Janpriya Kaur [1], Vibha Verma [1], Manmeet Kaur [1], Prabhjot Singh [1], Lovedeep Kaur [1], Gayatri Verma [3], Jagdish Singh [3], Ahmed Gaber [4] and Akbar Hossain [5]

[1]  Department of Soil Science, Punjab Agricultural University, Ludhiana 141027, India; sharmavivek@pau.edu (V.S.); janpriyakaur89@pau.edu (J.K.); vermavibha@pau.edu (V.V.); manmeetgill885@gmail.com (M.K.); prabh@pau.edu (P.S.); lovedeeppandher@pau.edu (L.K.)
[2]  Indian Institute of Soil Science, Bhopal 462038, India
[3]  Regional Research Station (PAU), Gurdaspur 143521, India; drgayatriverma@pau.edu (G.V.); jagdishsingh@pau.edu (J.S.)
[4]  Department of Biology, College of Science, Taif University, P.O. Box 11099, Taif 21944, Saudi Arabia; a.gaber@tu.edu.sa
[5]  Department of Agronomy, Bangladesh Wheat and Maize Research Institute, Dinajpur 5200, Bangladesh; akbarhossainwrc@gmail.com
*  Correspondence: ssdhaliwal@pau.edu (S.S.D.); arvindshukla2k3@yahoo.co.in (A.K.S.)

**Abstract:** The nutritional value of *Glycine max* L. (soybean) and its yield potential for improving sustainability of agricultural systems has resulted into its increased production. Soybean crop has potential to replace the rice crop in the rice-wheat cropping system. However, the crop has shown high sensitivity towards iron (Fe) deficiency, and thus recorded major yield and nutritional quality losses. Thus, a three-year field experiment was planned to compare the impact of the application rate (0.5% and 1.0%) and number of sprays of FeSO₄ on yield, Fe nutrition, and economic outcomes of soybeans. The Fe application posed a beneficial impact on the studied parameters due to an increase in enzymatic activity of Fe-containing enzymes. Among various treatments, maximum increase in grain and straw yield (3064 and 9341 kg ha⁻¹, respectively) was obtained with 0.5% FeSO₄ application at 30, 60, and 90 DAS over the control (2397 and 6894 kg ha⁻¹, respectively). Similar results were attained for grain Fe concentration (69.9 mg kg⁻¹) and Fe uptake in grain and straw (214 and 9088 g ha⁻¹, respectively). The results were statistically non-significant, with the treatment in which 0.5% FeSO₄ was applied at 30 and 60 DAS. The economic returns of soybean cultivation were also highest with 0.5% FeSO₄ application at 30, 60, and 90 DAS with highest benefit; the cost (3.02) followed by treatment in which 0.5% FeSO₄ was applied at 30 and 60 DAS. Thus, 0.5% FeSO₄ application at 30, 60, and 90 DAS can be recommended for soybeans grown on sandy loam soil followed by 0.5% FeSO₄ application at 30and 60 DAS to harness maximum yield, Fe concentration, and profitability.

**Keywords:** biofortification; foliar application; *Glycine max* L.; Fe uptake; Fe nutrition; economic analysis

## 1. Introduction

Globally, major public health issues that affected a major portion of the world's population have been found to be associated with micronutrient deficiencies [1,2]. Iron deficiency has been recognized as one of the key factors to the global burden of diseases, particularly in developing countries. Its deficiency mainly results in anemia, leading to functional impairments of the human body [3]. The deficiency in the human body is mainly associated with the consumption of food that is low in nutrient content. In the past few decades, Fe deficiency chlorosis has been identified as a chief nutritional disorder among the crops grown in calcareous soils, which leads to suppression in yield and quality losses

of crops [4,5]. As a key component of electron chains and a co-factor of various enzymes, the presence of Fe in a sufficient amount is mandatory in plants. In plants, the presence of Fe plays a crucial role in photosynthesis and chlorophyll synthesis [6]. Thus, sufficient Fe levels in agricultural crops are crucial to combat Fe deficiency. Too much Fe may also pose toxic effects on plant growth; thus it is mandatory to optimize the restricted availability of Fe to plants [7].

Soybeans (*Glycine max* L.), as a high protein source, can play a vital role in bridging the gap between nutrient intake and nutrients required by humans [8]. Moreover, its sustainable yield, economic returns, and contribution towards maintaining soil health increases its importance. It has a prominent place as an important seed legume, with a 25% contribution in the production of vegetable oil globally along with two-thirds of its protein concentrate used for livestock feeding [9]. India is the 4th-largest producer of soybeans in the world. This crop has huge potential for elevating farmers' economic status in many different regions of the country. The diverse uses of soybean include its intake as dal and soya milk, and also its role as an ingredient for bakery products [10]. Further, low saturated fat content with no cholesterol and presence of omega-3 fats along with minerals including calcium, magnesium, ferrous and selenium in ample amount make its consumption advantageous [11]. Thus, soybeans are considered as a beneficial food rich in protein and amino acids important for human body, which reduces the risk of various severe diseases including cancer, heart disease, and osteoporosis.

In Punjab, rice-wheat is a major cropping system and productivity of these crops are either stagnating or declining. The cultivation of rice crop has become less profitable and more dangerous than wheat for sustainable agriculture due to higher water demand. To maintain sustainable agriculture, there is a need to replace the rice crop with other crops. Soybean crop has large potential to fulfill the food and nutrition requirement by enhancing its productivity. The appearance of Fe deficiency in soybeans has been identified as a major reason in marring crop growth and takes a toll on productivity [12]. The distinctive yellow symptoms appear when the plant enters the 1st to 3rd trifoliate leaf stage and includes interveinal yellowing of younger leaves, whereas leaf veins remain green. Under severe deficiency of Fe, plant leaf edges become necrotic (turn brown) and this condition might end with the death of entire leaves or even plants [12]. These symptoms are mostly observed in irregularly shaped spots randomly distributed across a field.

Iron deficiency in soybeans appears not only due to the deficiency of available Fe in soil, but also due to soil conditions that prevent Fe uptake by soybean roots. In the Indo-Gangetic Plain of India, most of the soils are alkaline and low in available Fe content [13]. The excess rainfall during kharif season also causes a lack of oxygen around the roots, which inhibits the plant's capability to take up bioavailable Fe from soil. To avoid these losses, foliar application of Fe is a viable option to minimize the yield loss due to Fe deficiency in soybeans [14]. Previous reports have suggested that foliar application of Fe has improved the quantitative and qualitative production of various crops. For instance, yield and Fe concentration in chickpeas has significantly improved with foliar spray of Fe [5]. Likewise, Fe fertilization through foliage has resulted in a higher yield and grain Fe concentration in wheat crop [15]. Thus, the study was performed to optimize the number of applications and doses of application rates to enhance the yield, and Fe concentration in grain and economic outcomes of soybean cultivation.

## 2. Materials and Methods

### 2.1. Site Specification and Experimental Design

The experiment designed to study the impact of foliar application of $FeSO_4 \cdot 7H_2O$ on yield, Fe concentration, and economics of soybean cultivation was conducted for three years (2018, 2019, and 2020) at a research farm, Department of Soil Science, PAU Ludhiana (30°56' N, 75°52' E and 247 m above mean sea level). The soil was sandy loam with a pH of 7.26, EC = 0.38 dS m$^{-1}$, and the soil organic carbon was 0.34% [16,17]. The total N, available P, and K were 0.39%, 19.66 mg kg$^{-1}$, and 128.85 mg kg$^{-1}$, respectively [16,18,19].

The initial level of Zn, Cu, Fe, and Mn in soil were, respectively, 2.16, 0.80, 5.44, and 3.96 mg kg$^{-1}$ [20]. The experiment was laid out in a randomized block design with three replications. The study involved a total of seven treatments and details of treatment are given in Table 1. The weather parameters, i.e., monthly temperature and rainfall patterns, during the experimental period are given in Figure 1. The monthly average maximum and minimum temperature during the experimental period varied from 30.5 to 40.3 °C and 15.9 to 27.2 °C, respectively. The total rainfall during the crop period varied from 401 to 844 mm, which was highest in 2018 and lowest in 2020.

**Table 1.** Treatment details of experiment at research farm of PAU Ludhiana.

| Sr. No. | Treatment Detail | No. of Sprays | Days after Sowing (DAS) |
|---|---|---|---|
| T1 | Control | — | — |
| T2 | 0.5% FeSO$_4$·7H$_2$O | 1 | 30 |
| T3 | 0.5% FeSO$_4$·7H$_2$O | 2 | 30 and 60 |
| T4 | 0.5% FeSO$_4$·7H$_2$O | 3 | 30, 60 and 90 |
| T5 | 1.0% FeSO$_4$·7H$_2$O | 1 | 30 |
| T6 | 1.0% FeSO$_4$·7H$_2$O | 2 | 30 and 60 |
| T7 | 1.0% FeSO$_4$·7H$_2$O | 3 | 30, 60 and 90 |

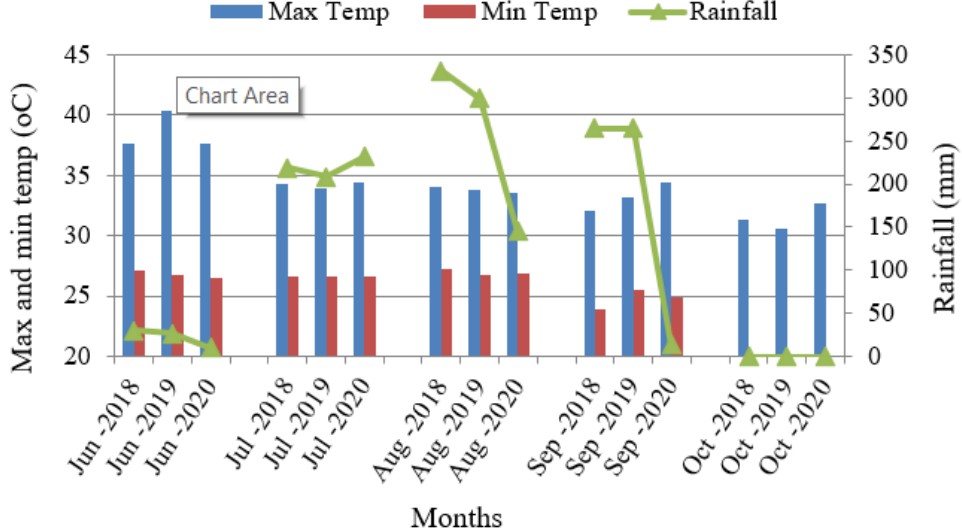

**Figure 1.** Monthly average maximum and minimum temperature as well as rainfall during the experimental period.

### 2.2. Fertilization and Management Practices

The field was plowed twice, followed by planking. A recommended dose of 31.2 kg of N (70 kg urea), 60 kg of P$_2$O$_5$ ha$^{-1}$ was applied as basal through urea and single superphosphate at the time of sowing. The soybean variety SL 958 was used for the experiment. The SL 958 is a recently recommended, high yielding variety for the Punjab state, having 41.7% protein and 20.2% oil content. It is also highly resistant to yellow mosaic virus and soybean mosaic virus. The soybean seed was treated with *Bradyrhizobium* sp. before sowing. The sowing was done in the second week of June via *pora* method with a 4–5 cm for plant spacing and row to row spacing was kept 45 cm.

### 2.3. Estimation of Yield and Fe Concentration

Plants were harvested manually in the last week of October followed by the collection of grain and straw samples for analysis. The samples were air-dried before drying in an

oven at 65 °C for 48 h to determine the dry weights of plant components. Oven-dried plant samples were further grounded to a fine material using a mechanical grinder. A representative grounded straw sample of 1.0 g and grain sample of 0.5 g were digested using a di-acid mixture ($HNO_3$:$HClO_4$: 3:1) on an electric hot plate [21]. The Fe concentration was estimated from the digested plant extracts through atomic absorption spectrophotometer Model AA 240 FS, Company Varian, Troisdorf, Germany. The Fe uptake in soybeans was calculated employing the following formula:

$$\text{Fe uptake in seed or straw } (\text{g ha}^{-1}) = \frac{\text{Yield } (\text{kg ha}^{-1}) \times \text{Concentration } (\text{mg kg}^{-1})}{10^3}$$

*2.4. Iron (Fe) Use Efficiency Indices and Economic Analysis*

The agronomic efficiency ($AE_{Fe}$) mobilization efficiency index ($ME_{Fe}$), physiological efficiency of Fe viz. ($PE_{Fe}$) and apparent recovery efficiency ($ARE_{Fe}$) of plants subjected to foliar Fe application were determined with the following equations [22]:

$$AE = \frac{Y_t - Y_c}{F}$$

$$ME = \frac{\text{Nutrient concentration in seed}}{\text{Nutrient concentration in straw}}$$

$$PE = \frac{Y_t - Y_c}{NU_t - NU_c}$$

$$ARE = \frac{NU_t - NU_c}{\text{Nutrient applied } \left(\text{kg ha}^{-1}\right)} \times 100$$

where $Y_t$ and $Y_c$ represent the seed yield (kg ha$^{-1}$) of soybeans in Fe-fertilized plants and in untreated plants, respectively; F represents the amount of fertilizer applied (kg ha$^{-1}$). $NU_t$ and $NU_c$ represent the total Fe uptake (kg ha$^{-1}$) of soybeans in Fe-fertilized plants and in untreated plants, respectively.

For economic analysis, the fertilizer cost is given in the United State Dollar (USD) ha$^{-1}$, for various treatments in the experiment worked out separately, considering the prevailing prices of fertilizers at the time of their application. Gross return (value of additional yield) was calculated based on the MSP (price for minimum support) of soybeans by the Indian government during the years of study. Net return (USD ha$^{-1}$) was obtained by subtracting the fertilizer cost from the gross return as given below [23].

$$\text{Net Return } (\text{USD ha}^{-1})$$
$$= \text{Gross return } (\text{USD ha}^{-1}) - \text{Cost of cultivation } (\text{USD ha}^{-1})$$

B:C ratio was calculated by using the following equation:

$$\text{B : C ratio} = \frac{\text{Gross return } \left(\text{USD ha}^{-1}\right)}{\text{Cost of cultivation } \left(\text{USD ha}^{-1}\right)}$$

*2.5. Statistical Analysis*

The data was analyzed statistically under ANOVA design using SPSS version 16.0 software. Duncan Multiple Range test was performed to test the least significant difference (LSD) using a probability level of $p \leq 0.05$ between the mean values of treatments.

## 3. Results

### 3.1. Impact of Foliar Application of $FeSO_4 \cdot 7H_2O$ on Yield

The variation in seed and straw yield of soybeans, as affected by foliar application of $FeSO_4 \cdot 7H_2O$ at different concentrations and different number of sprays are shown in Table 2. The results showed that foliar application of $FeSO_4 \cdot 7H_2O$ significantly enhanced the seed yield of soybeans. The mean of three-year data showed that the highest seed yield was recorded in treatment T4 (3064 kg ha$^{-1}$), which was statistically non-significant with treatment T3 (2801 kg ha$^{-1}$). The minimum seed yield was observed under treatment T1 (2397 kg ha$^{-1}$), which was statistically non-significant with treatment T2 (2619 kg ha$^{-1}$) and T7 (2669 kg ha$^{-1}$). Likewise, foliar application of $FeSO_4 \cdot 7H_2O$ enhanced the straw yield to a significant extent in treatments T3 (8229 kg ha$^{-1}$), T4 (9341 kg ha$^{-1}$), and T5 (8229 kg ha$^{-1}$) as compared to the control (6894 kg ha$^{-1}$).

**Table 2.** Impact of foliar application of $FeSO_4 \cdot 7H_2O$ on seed and straw yield of soybeans.

| Sr. No. | No. of Sprays | Seed Yield (kg ha$^{-1}$) | | | | Straw Yield (kg ha$^{-1}$) | | | |
|---|---|---|---|---|---|---|---|---|---|
| | | 2018 | 2019 | 2020 | Mean | 2018 | 2019 | 2020 | Mean |
| T1 | 0 | 2150 [c] | 2718 [b] | 2323 [d] | 2397 [c] | 5782 [b] | 7636 [e] | 7240 [b] | 6894 [c] |
| T2 | 1 | 2298 [bc] | 2965 [a] | 2619 [cd] | 2619 [bc] | 6079 [a] | 9217 [bcd] | 7586 [b] | 7636 [bc] |
| T3 | 2 | 2570 [a] | 3039 [a] | 2817 [bc] | 2801 [ab] | 6622 [ab] | 10,205 [ac] | 7833 [b] | 8229 [ab] |
| T4 | 3 | 2693 [a] | 3262 [a] | 3188 [a] | 3064 [a] | 7092 [a] | 11,441 [a] | 9489 [a] | 9341 [a] |
| T5 | 1 | 2372 [b] | 2965 [a] | 2817 [b] | 2718 [b] | 6301 [a] | 10,304 [ab] | 8080 [b] | 8229 [ab] |
| T6 | 2 | 2397 [b] | 2916 [ab] | 2916 [ab] | 2743 [b] | 6252 [a] | 10,107 [ad] | 8056 [b] | 8130 [abc] |
| T7 | 3 | 2273 [bc] | 2866 [b] | 2866 [a] | 2669 [bc] | 4300 [c] | 8945 [bcde] | 8945 [a] | 7388 [bc] |
| CD (*p* = 0.05) | — | 148 | 371 | 321 | 298 | 1137 | 1656 | 1038 | 1310 |

Values within a column succeeded by different small letters (a, b, c, d, e) differ significantly between different treatments at *p* < 0.05 significance level.

### 3.2. Impact of Foliar Application of $FeSO_4 \cdot 7H_2O$ on Fe Concentration

The results of Fe concentration in the seed and straw of soybeans with treatment variations are presented in Table 3. Foliar application of $FeSO_4 \cdot 7H_2O$ significantly enhanced the Fe concentration in the seed and straw of soybeans as compared to the control. In soybean seeds, Fe concentration showed unexpected trends, as maximum value was attained in treatment T4 (69.9 mg kg$^{-1}$). The Fe concentration in treatment T4 was statistically non-significant with treatment T3 (67.9 mg kg$^{-1}$). Thus, maximum Fe concentration in grains can be obtained by two applications of $FeSO_4$ applied at 0.5%. In the case of straw Fe concentration, the highest values were obtained in treatment T6 (1040 mg kg$^{-1}$), which were not statistically different with treatment T6 (1037 mg kg$^{-1}$). Thus, Fe concentration in straw increased with the increase in application rate of $FeSO_4 \cdot 7H_2O$ from 0.5% to 1% as well as with an increase in the number of applications from 1 to 3. The lowest Fe concentration in grain and straw was observed in treatment T1 (53.9 and 865 mg kg$^{-1}$), which was significantly lower than other treatments.

### 3.3. Impact of Foliar Application of $FeSO_4 \cdot 7H_2O$ on Fe Uptake

The three-year results with different levels of Fe and number of foliar sprays for Fe uptake by seed and straw of soybean are represented in Table 4. The observed trend showed that foliar application of $FeSO_4 \cdot 7H_2O$ significantly enhanced the Fe uptake in seed and straw over the control, as results of treatment T1 are significantly lower than other treatments. The highest Fe uptake in seed and straw was recorded in treatment T4 (214 and 9088 g ha$^{-1}$, respectively) and control (129 and 5961 g ha$^{-1}$, respectively). The results of treatment T4 for seed Fe uptake were statistically non-significant with treatment T3 (190 g ha$^{-1}$).

**Table 3.** Impact of foliar application of $FeSO_4 \cdot 7H_2O$ on Fe concentration of soybeans.

| Sr. No. | No. of Sprays | Seed Fe Concentration (mg kg$^{-1}$) | | | | Straw Fe Concentration (mg kg$^{-1}$) | | | |
|---|---|---|---|---|---|---|---|---|---|
| | | 2018 | 2019 | 2020 | Mean | 2018 | 2019 | 2020 | Mean |
| T1 | 0 | 45.4 [d] | 48.3 [e] | 59.5 [c] | 53.9 [d] | 899 [c] | 921 [bc] | 774 [d] | 865 [d] |
| T2 | 1 | 56.8 [c] | 56.8 [cd] | 65.2 [b] | 59.6 [c] | 955 [b] | 960 [a] | 840 [cd] | 918 [c] |
| T3 | 2 | 65.6 [a] | 67.9 [ab] | 70.3 [a] | 67.9 [ab] | 1020 [a] | 962 [a] | 972 [b] | 985 [b] |
| T4 | 3 | 69.5 [a] | 71.3 [a] | 69.0 [a] | 69.9 [a] | 1027 [a] | 957 [a] | 935 [b] | 973 [b] |
| T5 | 1 | 61.8 [bc] | 64.8 [a] | 65.7 [b] | 64.1 [bc] | 986 [ab] | 949 [ab] | 909 [bc] | 948 [bc] |
| T6 | 2 | 64.6 [a] | 63.6 [bc] | 64.7 [b] | 64.3 [b] | 1018 [a] | 960 [a] | 1133 [a] | 1037 [a] |
| T7 | 3 | 65.7 [ab] | 55.7 [de] | 68.8 [a] | 63.4 [bc] | 1027 [a] | 947 [ac] | 1146 [a] | 1040 [a] |
| CD ($p = 0.05$) | — | 6.22 | 7.4 | 2.98 | 5.5 | 51 | 32 | 71.2 | 51 |

Values within a column succeeded by different small letters (a, b, c, d, e) differ significantly between different treatments at $p < 0.05$ significance level.

**Table 4.** Impact of foliar application of $FeSO_4 \cdot 7H_2O$ on Fe uptake of soybean.

| Sr. No. | No. of Sprays | Seed Fe Uptake (g ha$^{-1}$) | | | | Straw Fe Uptake (g ha$^{-1}$) | | | |
|---|---|---|---|---|---|---|---|---|---|
| | | 2018 | 2019 | 2020 | Mean | 2018 | 2019 | 2020 | Mean |
| T1 | 0 | 98 [e] | 131 [d] | 138 [d] | 129 [d] | 5198 [e] | 7032 [d] | 5604 [g] | 5961 [f] |
| T2 | 1 | 131 [d] | 168 [c] | 171 [c] | 156 [c] | 5805 [d] | 8848 [c] | 6372 [f] | 7012 [e] |
| T3 | 2 | 169 [ab] | 206 [ab] | 198 [b] | 190 [a] | 6755 [b] | 9848 [b] | 7614 [d] | 8111 [c] |
| T4 | 3 | 187 [a] | 233 [a] | 220 [a] | 214 [a] | 7283 [a] | 10,949 [a] | 8872 [c] | 9088 [a] |
| T5 | 1 | 147 [cd] | 192 [bc] | 185 [bc] | 174 [bc] | 6213 [c] | 9779 [b] | 7345 [e] | 7801 [d] |
| T6 | 2 | 155 [bc] | 185 [bc] | 189 [bc] | 176 [bc] | 6364 [c] | 9672 [b] | 9127 [b] | 8422 [b] |
| T7 | 3 | 149 [bcd] | 160 [cd] | 197 [b] | 169 [bc] | 4416 [f] | 8471 [c] | 10,251 [a] | 7684 [d] |
| CD ($p = 0.05$) | — | 21 | 32 | 18 | 24 | 258 | 453 | 174 | 302 |

Values within a column succeeded by different small letters (a, b, c, d, e, f, g) differ significantly between different treatments at $p < 0.05$ significance level.

### 3.4. Impact of Foliar Application of $FeSO_4 \cdot 7H_2O$ on Efficiency Indices and Economic Outcomes

The Fe use efficiency indices of soybeans with foliar applications of $FeSO_4 \cdot 7H_2O$ were estimated for different treatments (Table 5). The AE and PE maximum was recorded in T4 treatment with values 178(kg kg$^{-1}$) and 0.97(kg g$^{-1}$). The ME was higher in treatment T3, T4, and T5 (0.7), whereas ARE was maximum in treatment T3 (88.4%) followed by T2 (86.2%) and T4 (85.7%).

**Table 5.** Impact of foliar application of $FeSO_4 \cdot 7H_2O$ on Fe use efficiency indices of soybean.

| Treatments | AE$_{Fe}$ (kg kg$^{-1}$) | ME$_{Fe}$ | PE$_{Fe}$ (kg g$^{-1}$) | ARE$_{Fe}$ (%) |
|---|---|---|---|---|
| T1 | — | 0.06 | — | — |
| T2 | 177 | 0.06 | 0.89 | 86.2 |
| T3 | 161 | 0.07 | 0.78 | 88.4 |
| T4 | 178 | 0.07 | 0.97 | 85.7 |
| T5 | 128 | 0.07 | 0.88 | 75.4 |
| T6 | 69 | 0.06 | 0.63 | 50.2 |
| T7 | 36 | 0.06 | 0.43 | 23.5 |

The results of economic analysis parameters, including on cost of cultivation, net return, and B:C of soybeans, are represented in Figure 2. The data demonstrated that cost of cultivation was highest in treatment T7 ($864) and lowest in treatment T1 ($744). On the other hand, net return and B:C ratio was highest in treatment T4 ($1439 and 2.77, respectively) and lowest in treatment T1 ($1017 and 2.37, respectively).

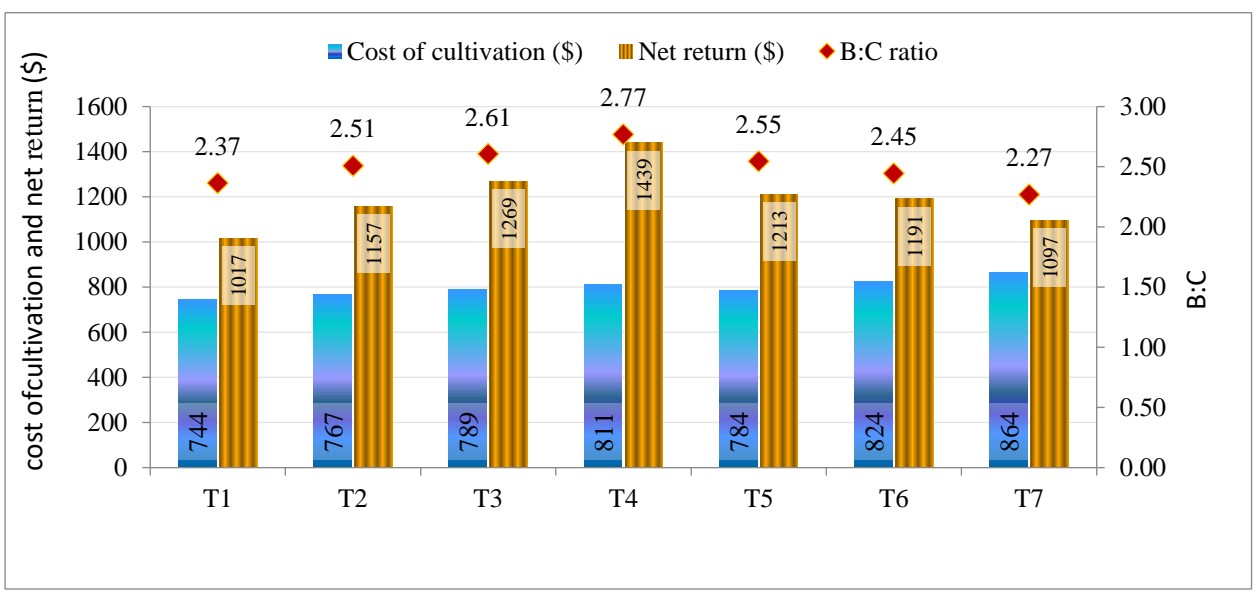

**Figure 2.** Impact of foliar application of $FeSO_4 \cdot 7H_2O$ on cost of cultivation, net return, and B:C.

## 4. Discussion

### 4.1. Impact of Foliar Application of $FeSO_4 \cdot 7H_2O$ on Yield

Earlier reports suggested that foliar application of $FeSO_4 \cdot 7H_2O$ not only improved the Fe absorption through foliage, but also increased the translocation with the plant. The increase in Fe translocation improved the seed and straw yield of soybeans. The Fe present in ferrodoxin and cytochrome structures acts as an electron carrier, thus playing a crucial role in various metabolic processes, including photosynthesis, chlorophyll construction, respiration, nitrogen fixation, DNA synthesis, and hormone production [23]. The photosynthesis efficiency and the functioning of the photosynthetic apparatus are strongly dependent on iron availability to crops. In the presence of insufficient iron, chlorophyll synthesis in the leaves suppresses, which reduces the vegetative growth and thus yield of the plant. Thus, significant increase in the biological yield of soybeans might be attributed to the improved leaf and stem nutrition and intensification of photosynthesis due to foliar application of Fe [24]. Moreover, the higher availability of Fe increased the enzymatic activities of Fe-containing enzymes has enhanced soybean yield. The results of the present study are concordant with the results obtained from [21], where foliar application of Fe increased the herbage yield of teosinte. Likewise, the yield of various other crops including maize, wheat, and rice has been found to increase with foliar application of Fe [25–27].

### 4.2. Impact of Foliar Application of $FeSO_4 \cdot 7H_2O$ on Fe Concentration

The trend recorded for Fe concentration in seeds revealed that a maximum increase in Fe concentration can be achieved through either two or three sprays of 0.5% $FeSO_4 \cdot 7H_2O$. The foliar spray at higher concentration (1.0% $FeSO_4 \cdot 7H_2O$) yielded lower Fe concentration in seeds, which was contrary to the results obtained for straw Fe concentration in soybeans. The results can be explained by the fact that Fe absorption in plant leaves is accompanied by its translocation in the plant body [23]. Out of the amount absorbed by the plant, an optimum amount of nutrients is translocated in the seed and the rest of the nutrients remains in leaves. Thus, application of a higher concentration of metal salt resulted in lower Fe concentration in seeds and higher Fe concentration in straw. Similar results were

obtained in previous reported studies for different crops including maize, wheat, and rice, where foliar application of Fe increased the Fe concentration in seed and straw [25–27].

### 4.3. Impact of Foliar Application of $FeSO_4 \cdot 7H_2O$ on Fe Uptake

The trend of Fe uptake in seed and straw was attributed to the combined effect of yield and Fe concentration. The exogenous supply of Fe enhanced the Fe bioavailability to soybeans at vegetative and reproductive phases due to more absorption of Fe through plant foliage [28]. Thus, Fe absorption enhanced the enzymatic activities, which improved the yield and Fe concentration of soybeans and thus enhanced the Fe uptake. Similar results of higher Fe uptake in plants with foliar Fe application have been previously reported [29,30]. At a higher application rate, Fe uptake decreased, which might be due to the toxic effects of Fe at a higher concentration.

### 4.4. Impact of Foliar Application of $FeSO_4 \cdot 7H_2O$ on Efficiency Indices and Economic Outcomes

The highest value of AE in treatment T4 indicated that soybean response in terms of yield was highest with three foliar sprays of $FeSO_4 \cdot 7H_2O$ (0.5%). The results of ME demonstrated that the application of $FeSO_4 \cdot 7H_2O$ at higher rate (1.0%) results in more Fe translocation towards the straw as compared to the seed. The lower values of ME also suggested more Fe translocation in straw as compared to the seeds. The PE results indicate the increase in yield per units of absorbed Fe by plants and identifies the role of Fe in increasing the soybean yield. The highest three foliar sprays of $FeSO_4 \cdot 7H_2O$ (0.5%) result in the maximum increase in yield with respect to Fe absorbed by the plant. The ARE value was highest under treatment T3, which showed that maximum Fe absorption by the plant was achieved with two sprays of $FeSO_4 \cdot 7H_2O$ @ 0.5%. Similar results of increase in efficiency indices with Fe application were reported in chickpeas [31].

Under economic analysis, the maximum cost of cultivation in treatment T7 was due to a higher number and higher rate of Fe applications over the other treatments. The highest net return and B:C ratio in treatment T4 was associated with the maximum yield. Thus, Fe application has proved beneficial for soybean cultivation. Similar results have been observed with foliar application of Fe in chickpeas [31].

### 5. Conclusions

The present study identified the beneficial role of Fein soybean (*Glycine max* L.) cultivation. Moreover, to achieve maximum yield and nutrition, it is mandatory to optimize frequency and rate of application of Fe through $FeSO_4 \cdot 7H_2O$. $FeSO_4 \cdot 7H_2O$ application significantly improved the yield and Fe concentration over the untreated plants, irrespective of application rate and number of sprays. However, application of 0.5% $FeSO_4 \cdot 7H_2O$ at 30, 60, and 90 DAS resulted in the highest increase in seed and straw yield followed by the treatment in which 0.5% $FeSO_4 \cdot 7H_2O$ was applied at 30 and 60 DAS. A similar trend was observed for Fe concentration in seeds and Fe uptake in seed and straw. A higher application rate resulted in the translocation of Fe into straw and not in edible seeds. The maximum economic profit in soybean cultivation was also recorded with 0.5% $FeSO_4 \cdot 7H_2O$ application at 30, 60, and 90 DAS. Thus, application of 0.5% $FeSO_4 \cdot 7H_2O$ at 30, 60, and 90 DAS can be suggested for soybean cultivation on sandy loam soil followed by 0.5% $FeSO_4 \cdot 7H_2O$ application at 30 and 60 DAS for sustainable yield and Fe nutrition.

**Author Contributions:** S.S.D., J.S., J.K., M.K., V.V., V.S., P.S. and L.K.; methodology, S.S.D., J.S., J.K.,V.S., V.V., M.K. and L.K.; software, S.S.D., G.V., P.S. and A.H.; validation, S.S.D., J.S., V.S., V.V., M.K., P.S. and L.K.; formal analysis, S.S.D. and A.H.; investigation, S.S.D., J.S., V.S. and L.K.; resources, M.K.; data curation, S.S.D., G.V., P.S. and A.H.; writing—original draft preparation, S.S.D., V.S. and A.K.S.; writing—review and editing, A.G., A.K.S. and A.H.; visualization, S.S.D., J.S., V.S. and L.K.; supervision, S.S.D., V.S., A.K.S. and V.V.; project administration, S.S.D., A.G., A.K.S. and A.H.; funding acquisition, A.G., A.K.S. and A.H. All authors have read and agreed to the published version of the manuscript.

**Funding:** This research was funded by the Department of Soil Science, Punjab Agricultural University, Ludhiana, India and ICAR-Indian Institute of Soil Science, Bhopal, 462038, Madhya Pradesh, India. This research was also partially funded by the Taif University Researchers for funding this research with Supporting Project number (TURSP-2020/39), Taif University, Taif, Saudi Arabia.

**Institutional Review Board Statement:** Not applicable.

**Informed Consent Statement:** Not applicable.

**Data Availability Statement:** The data will be provided as per demand.

**Acknowledgments:** The authors are thankful to the Indian Council for Agricultural Research (ICAR), Indian Institute of Soil Science (IISS), Bhopal, and the Taif University Researchers for funding this research with Supporting Project number (TURSP-2020/39), Taif University, Taif, Saudi Arabia for supporting to conduct the field trial.

**Conflicts of Interest:** The authors have no conflict of interest.

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
