# Peer review of "Biofortification of Soybean (Glycine max L.) through FeSO4·7H2O to Enhance Yield, Iron Nutrition and Economic Outcomes in Sandy Loam Soils of India"

_agriculture, doi:10.3390/agriculture12050586_

Round 1
Reviewer 1 Report
Comments to the Author
Abstract
The abstract is well written and meets the criteria for Agiculture. A brief describe about the application prospect of the research is suggested in this section.
Introduction
More details about soybean production in Punjab should be provided in this section. Is Punjab a major soybean producer in India? Are there other parts of India where the soybean production is affected by the deficiency of Fe. These backgrounds are of great benefit to explain the significance and application prospect of this study.
Material and Methods
Only one variety, SL 958, is used in this study. It is necessary to describe why this variety is used in the material method. Is it a representative local variety or a variety sensitive to Fe?
Results and Discussion
Why is the cost of T4 (0.5% FeSO4 application at 30, 60 and 90 DAS) significantly lower than that of T2 and T3 in Fig.1? It is the same as T1. Is this a mistake or is there any reason for it? The author should give an explanation about it, for this data is important for part of the conclusion in this research.
The significance and application prospects of this technique for soybean cultivation in other parts of the world should be also appropriately described in this section.
Author Response
All needful done on the body of text of manuscript under track changes mode.

Reviewer 2 Report
Dear authors,
The reviewed manuscript presents an interesting approach to the topic of biofortification of Fe soybean. Notes for the work are included in the comments.

Author Response
All needful done on the text of manuscript under track change mode.
